# Autonomous Shuttle Operating on Highways and Gravel Roads in Rural America: A Demonstration Study

**DOI:** 10.3390/geriatrics7060140

**Published:** 2022-12-08

**Authors:** Justin Mason, Cher Carney, John Gaspar

**Affiliations:** National Advanced Driving Simulator, College of Engineering, University of Iowa, Iowa City, IA 52242, USA

**Keywords:** older adults, rural transit, autonomous vehicles, community mobility

## Abstract

The safe integration of Automated Driving Systems (ADS) into the nation’s on-road transportation system, particularly in rural areas, could vastly improve overall quality of life for a rapidly growing segment of the US population. This paper describes findings from the first half (i.e., three of six phases) of a demonstration project called “ADS for Rural America”. The goal of this project is to conduct a series of demonstrations that utilizes an autonomous shuttle to show how older adults (≥65 years old) could be transported from their rural homes to other locations in rural areas, as well as an urban center. This paper examines older adults’ perceptions of automation before and after riding in an autonomous shuttle and their ratings of anxiety throughout the ride as they experience particular road types and maneuvers. After riding in the shuttle, older adults expressed decreased suspicion, increased trust, and increased reliability of ADS compared to baseline. Older adults reported low levels of anxiety during the 90 min ride in the shuttle. To promote the adoption and acceptance of ADS, older adults should be exposed to this technology.

## 1. Introduction

A relatively large and growing portion of rural and small-town residents are older Americans [1]. It is likely that as older Americans retire, they may seek out communities that offer affordable housing, small town quality of life, and are located within a relatively short drive of larger metropolitan areas. Despite community mobility obstacles (i.e., accessibility and availability of transportation options, driving cessation), many rural-dwelling older adults report rural benefits that include attachment to community, familiarity of their environment, and social participation [2]. This trend of an aging rural America is likely to continue as many older Americans desire to “age in place,” meaning they want to continue to live in their own home independently, regardless of age, income, or ability level [3]. Aging in place often outweighs the lack of transportation and local health care facilities in rural areas [2].

To help older adults keep this desired level of independence, more safe mobility options in all areas of the nation will need to exist. Approximately 40% of the rural US population has no public transportation at all and another 25% only has minimal service [4]. Additionally, those public and private transportation options that exist largely focus on getting older adults to and from medical appointments. However, healthy aging requires a holistic context of health, recognizing that factors like access to healthy food sources, community mobility, and social interaction also contribute to a positive quality of life.

The safe integration of Automated Driving Systems (ADS) into the nation’s transportation system, particularly in rural areas, could vastly improve overall quality of life for a rapidly growing segment of the US population. ADS and other public transport initiatives will need to cover transit gaps not currently served by the private sector (i.e., ridehailing services). It may be a few decades until the private sector is able to provide for rural areas across the US. Several studies and demonstration projects are looking into the feasibility of these types of services and the acceptance of this technology by this population [5,6,7,8,9,10,11,12]. However, this type of research is often done via survey without exposure to ADS. Individuals should have lived experiences riding in an autonomous vehicle (AV) to inform their perceptions of this emerging technology [5,6,13].

Requirements of people with different needs in society, such as adults with cognitive or physical impairments, must be addressed. While not all older adults have impairments, they are susceptible to age-related cognitive, sensory, or mobility declines that negatively affect their community mobility. Older adults and persons with disabilities are often not included in AV development and demonstration projects [14]. We need to have a much wider approach to research and development if we are to develop accessible transportation modes. Although sparse, researchers have found that exposure to autonomous shuttles, positively influences older adults’ perceptions of AVs. However, these studies have occurred in controlled suburban areas with shuttles operating at slow speeds (<10 mph), during ideal weather conditions (i.e., sunny without heavy rain), with rides lasting roughly 15 min [5,6,12]. Findings from these studies provide a strong foundation that will be expanded upon in this study. Specifically, older adults will ride in an AV for about an hour, with speed ranging from 15 to 65 mph, on surfaced and gravel roads, with the potential to experience inclement weather.

This paper describes three phases of a demonstration project called “Automated Driving Systems for Rural America” (https://adsforruralamerica.uiowa.edu/, accessed on 30 July 2022). The goal of this project is to conduct a series of demonstrations that utilizes an autonomous shuttle to show how transportation-challenged populations, like older Americans, could be transported from their rural homes to other locations in rural areas, as well as an urban center. Data is being collected regarding the state of the automation, the vehicle performance, as well as the perceptions of the riders. During each new phase, the project team is also assessing the automation’s performance and using the data collected to inform improvements in successive phases.

The project is comprised of six phases; however this paper will focus on the three phases completed thus far. The automation capability, defined as the percentage of the route driven by the ADS, of the shuttle was enhanced after each phase. Specifically, the percentage of the route driven in automation per phase increased from 58% in phase 1 to 93% for phase 3. The rationale for studying the first three phases is these phases had the greatest magnitude of change in automation (percentage of drive using automation) and we will explore whether the percentage of automation influenced older adults’ perceptions of the technology. This paper focuses on the: (a) riders’ perception of higher levels of automation both pre- and post-trip; and (b) ratings of anxiety during the shuttle ride as well as around particular road types and maneuvers.

## 2. Materials and Methods

Adults over the age of 65 were recruited to ride inside the AV. Eligible participants were: (a) able to sit for up to 3 h at a time; (b) felt comfortable riding in the shuttle with strangers; and (c) able to read English. Participants were not eligible to participate in the study if they were: (a) having difficulties with memory or confusion; (b) diagnosed with dementia, delirium, mild cognitive impairment, or other severe neurological impairments (i.e., Parkinson’s, Huntington’s); (c) diagnosed with an anxiety disorder accompanied with an acute anxiety or a panic attack in the last 6 months; (d) diagnosed with seizures, narcolepsy, or epilepsy with an episode in the past 12 months; (e) deaf in both ears; or f) prone to motion sickness. They were recruited from the communities along the route using flyers and word of mouth as well as through emails sent through the National Advanced Driving Simulator registry. Recruitment emails were sent to 701 older adults in the registry. 235 adults completed an online survey from the registry, flyers, or word of mouth. Of the 235 adults, 189 individuals met the inclusion and exclusion criteria. 105 participants were called sequentially and 85 were scheduled to participate in the study. 20 of the 105 adults had scheduling conflicts and expressed interest in participating in a future phase of the project.

### 2.1. Vehicle

The ADS for Rural America project shuttle is a custom-built, mobility-friendly Ford Starlite Transit. This accessible vehicle is outfitted with a wheelchair lift, securement location, and securement system. The vehicle is based on a 2020 model year Ford Transit 350 HD Cutaway Cab chassis with a 138” wheelbase (Figure 1). The interior cab has two forward-facing seats, one for the safety driver and another for the co-pilot. Behind the bulkhead, the rear driver’s side has two rows of two seats and a built-in wheelchair restraint for participants while the passenger’s side has a single seat that is reserved for the research and data collection staff. To support accessibility needs of older adults, the vehicle has storage racks for luggage or groceries and a low first entry step, deep step wells, and a ramp that is compliant with the Americans with Disabilities Act (ADA) and Rehabilitation Act.

### 2.2. Route

ADS for Rural America takes place in Eastern Iowa, traveling from Iowa City through various small communities and rural areas south of the city. These communities and rural areas were selected to highlight how ADS can provide significant public benefits to quickly growing transportation-challenged populations in the US that will likely receive insufficient private sector investment in the near future. The route lasted ≈90 min and consisted of four stops (i.e., starting locations based on randomization) which included The Kolona Library, The Hills Senior Center, The Riverside Casino, and the Iowa City Marketplace (see Figure 2a). Older adults started and ended their ride at the same location and did not exit the vehicle during any of the stops. These locations were chosen because they were ones that people might be interested in traveling to. It was also important that the route incorporate as many different types of roadways and parking “types” as possible. Specifically, the route was designed to include gravel, highway, slow speed rural roads, and roads utilized by pedestrians, cyclists, farm equipment, and horse and buggy (i.e., Amish community in Kolona). During the ride, a display mounted at the front of the shuttle provided the rider with the location of the vehicle along the route as well as the state of the automation (see Figure 2b). Each older adult was also provided with a Microsoft Surface Go 2 tablet that had several apps for them to entertain themselves with during the ride as well as the current location of the vehicle along the route. The shuttle contained a safety driver, co-pilot, researcher, and a maximum of two participants.

The same route was used throughout all phases of this study, with automation capability (percentage of drive) increasing in each phase of the project. For example, in Phase 1 the vehicle was driven under automation primarily on controlled access roadways (interstates) and stretches of rural highways, and on/off ramps were added in Phase 2. By Phase 3, the vehicle was able to drive under automation through the urban areas, read and respond to the color of traffic lights (via cameras), and navigate four-way stops. The four starting locations were randomly assigned to each participant as were the time of day that the drives occurred in order to expose the vehicle to various types and levels of mixed traffic and varying lighting conditions. During the drives, participants were informed when the shuttle was operating with automation or via manual control of the safety driver via a central information display located at the front of the shuttle.

### 2.3. Surveys

Demographics and Transportation Questionnaire. The questionnaire collected participant responses for age, gender, education, income, impairments/disabilities, annual driving history, driver’s license restrictions, and their preferences related to driving and transportation. They were also asked about the types of technology (e.g., adaptive cruise control, lane keep assist) on their vehicle.

Perceptions of Highly Autonomous Vehicles Survey. While riding in the shuttle, occupants were asked to complete both a pre- and post-ride survey regarding their trust and acceptance of highly autonomous vehicles (HAVs). This type of vehicle was defined as one that is “capable of driving on its own in some situations but is aware of its limitations and calls for the driver to take over when necessary.” Survey questions were modified from previous technology acceptance models and AV questionnaires [15,16,17,18,19,20] written to expressly get at the following constructs: (a) situational trust; (b) safety; (c) community mobility; (d) hesitation; (e) trust and reliability; and (g) intention to use. Item responses were Likert-scaled ranging from 1 to 5 (strongly disagree to strongly agree). The survey contained 34 items but only 14 items were analyzed in this study as they were particularly relevant to the automation capabilities that changed throughout these specific phases (i.e., Phases 1 through 3). Furthermore, only a portion of the items were analyzed to limit the potential of type I error.

Anxiety Rating Questionnaire. Participants were also asked to provide a rating of their anxiety level from 0 to 10, with 0 being “not at all anxious.” These ratings were given at nine specific locations along the drive and were the same for each participant, although they did vary in the order they were given depending on the starting location for the drive. Figure 2 is a map showing where each of these ratings occur along the drive. A pre-drive anxiety rating was obtained for everyone before the drive began. Ratings were recorded at the following locations (see Figure 2a):Hwy 6 in Iowa CityMerge onto Hwy 218Turn onto Hwy 22Business district of RiversideDowntown KalonaHwy 1 ruralGravel roadUnmarked blacktop roadHwy 1 intersection

Data from the three phases of this project were collected in Qualtrics via a tablet and collated in RStudio using R. Descriptive statistics are displayed for older adults’ demographics, self-reported transportation habits, and perceptions of HAVs. Continuous data are displayed as mean and standard deviation. Categorical data are displayed as count (n) and percentage. Data that violated parametric assumptions are displayed as median and interquartile range (IQR). Prior to analysis, the dependent variables were screened for normality (i.e., skewness >2, kurtosis >9, Shapiro–Wilk *p* < 0.05, and Q-Q plot observation). Normality violations are reported in the results section and informed post hoc test selection. A series of two-way mixed ANOVAs were conducted for the perceptions of HAVs to explore the time effect (pre vs. post), group effect (phase), and group by time interaction. A series of two-way mixed ANOVAs were conducted for anxiety ratings to explore the location effect (9 locations of the trip), group effect (phase), and group by location interaction. A repeated measures ANOVA was conducted across all phases to assess the effects of trip time on anxiety. Multiple comparisons were controlled for using the Benjamini-Hochberg procedure [21].

## 3. Results

The older adults (N = 85) were well-educated (83.5% reported having at least an undergraduate degree) with all but one participant maintaining an active driver’s license. Driving was their primary mode of transportation and an overwhelming majority reported not using public transit (98.8%), paratransit (100%), taxis (96.5%), shuttles (95.3%), or ridehailing services (94.2%; i.e., Uber). The primary reasons for not using other modes of transportation were the accessibility and availability of these services. Demographic and transportation results are displayed by phase in Table 1.

Older adults’ perceptions were not normally distributed as indicated by the Shapiro–Wilk test (*p*’s < 0.05). The two-way mixed ANOVA displayed a time effect (pre vs. post) for older adults’ intention to use HAVs (*F*(1,162) = 5.24, *p* = 0.023), reliability of HAVs (*F*(1,163) = 7.85, *p* = 0.006), suspicion of HAVs (*F*(1,163) = 9.38, *p* = 0.003), trust in HAVs (*F*(1,162) = 5.66, *p* = 0.019), and situational trust while HAVs operated on the highway (*F*(1,163) = 10.3, *p* = 0.002). When controlling for multiple comparisons using the Benjamini-Hochberg procedure, intention to use HAVs (*p* = 0.121) was no longer significant. After riding in the shuttle, participants reported decreased suspicion of HAVs and increased trust in HAVs, increased situational trust while HAVs operated on the highway, and increased reliability of HAVs compared to baseline. Ratings of trust in HAVs and trust in HAVs on the highway are displayed in Figure 3 by time and phase.

A group effect (i.e., between phases) was observed from the two-way mixed ANOVA for HAVs interacting with pedestrians and bicyclists (*F*(2,163) = 6.52, *p* = 0.002), HAVs responding to lights/signs (*F*(1,163) = 4.21, *p* = 0.017), and feeling safe riding in an HAV (*F*(1,163) = 5.40, *p* = 0.005). When controlling for multiple comparisons using the Benjamini-Hochberg procedure, responding to traffic lights/signs (*p* = 0.114) was no longer significant. Comparing for multiple comparisons, post hoc analyses using Wilcoxon rank sum tests displayed higher levels of safety in phases 2 (*p* = 0.012) and 3 (*p* = 0.002) compared to phase 1 and decreased concerns about HAVs interaction with pedestrians and cyclists in phase 3 compared to phases 1 (*p* = 0.004) and 2 (*p* = 0.025). No time by group interaction effects were observed. Item responses are displayed as proportion by phase and time in Figure 4 and descriptive statistics are displayed in Table 2.

Anxiety ratings were not normally distributed as indicated by the Shapiro–Wilk test (*p*’s < 0.05). The repeated measures ANOVA did not reveal a difference for anxiety across time points (*F*(9,771) = 0.271, *p* =0.982). Since the length of the ride (i.e., time), did not affect older adults’ anxiety ratings, the location was further explored without considering time as a covariate.

The two-way mixed ANOVA for anxiety revealed a group effect between the phases, *F*(2,751) = 9.055, *p* < 0.001. The post hoc analyses using Wilcoxon rank sum tests displayed no differences for self-reported anxiety ratings between the phases. However, no effects were observed for location (*p* = 0.981) or group (i.e., phase) by location interaction (*p* = 0.991). Anxiety ratings are displayed by route location between phases (Figure 5). Individual ratings of anxiety are also displayed within Figure 5. Descriptively, older adults in Phase 3 reported higher ratings of anxiety. The boxplot in Figure 5 displays older adults’ anxiety ratings by location as well as the distribution of individuals’ anxiety ratings.

## 4. Discussion

This is the first demonstration project that has exposed individuals to a high-speed autonomous shuttle operating on highways and gravel roads in rural America. On average, older adults reported low levels of anxiety during their 90 min ride in the autonomous shuttle. Although there were no significant differences for anxiety between the phases, Figure 5 displays that a few individuals expressed increased anxiety in Phase 3 (95% automation) while merging on the highway or at highway intersections. This suggests that certain older adults may be especially anxious about autonomous shuttles operating on the highway or while merging with high-speed traffic. In the next three phases of this project, special emphasis will be put on these traffic maneuvers to gain insight as to why individuals do not feel comfortable with the shuttle operating via automation and if driving performance varies between the shuttle operating in automation or via manual control of the operator.

After riding in the shuttle, older adults expressed decreased suspicions, increased trust, and felt that HAVs were more reliable compared to their initial understanding of HAVs at baseline. These findings contribute to the literature and align with other studies that exposed older adults to autonomous shuttles operating on paved roads at slower speeds (i.e., 10 mph) [5,12]. Although there were very few differences between the groups (i.e., phases), older adults that rode in the shuttle while it was operating at higher automation capability (95%) reported feeling safer and had less concerns of the shuttle operating near pedestrians and cyclists compared to those that experienced lower automation capabilities (i.e., Phase 1 experienced 58% automation). Older adults may have supervised the state of the automation which was displayed in the vehicle and on their tablet throughout the drive. We also postulate that the safety operator may have influenced their perceptions as the operators were also becoming more familiar with the automation during each drive and throughout the phases, although the automation capabilities were also evolving throughout the phases. Interestingly, older adults across all three phases underestimated the automation capabilities of the vehicle (% of the drive spent in automation). This may have occurred due to takeovers performed by the safety operator and will be explored in the last three phases of this project. Ultimately, experience with this technology, as it improves and evolves, will likely promote acceptance and adoption practices of automation.

The primary study limitation is generalizability due to sampling bias. The study likely contained older adults that were particularly interested in ADS and did not include those that were that were hesitant, resistant, or reluctant to ride an autonomous shuttle operating in rural America. To overcome sampling bias, future phases of this project may target older adults that are reluctant to ride in an autonomous shuttle, persons with disabilities that require assistance with ingress or egress, or those with weak mental models (i.e., understanding) of automation. The presence of a safety operator in the driving seat likely influenced older adults’ perceptions while riding in the shuttle. In the US, the National Highway Traffic Safety Administration approves the proposed shuttle route and requires safety operators to always have their hands on the vehicle controls. This often looks different in the slow-speed shuttles (e.g., Navya Autonom or EasyMile EZ10), which use a remote controller resembling an Xbox controller or drone remote controller. Results from future demonstration projects may elucidate the effects of changes to regulation, policy, and availability of AVs on older adults’ perceptions of HAVs. Demonstration projects should continue to target special populations that have the most to gain from adopting ADS and focus on specific driving maneuvers that are perceived as risky or complex (i.e., merging at high speeds).

Although nationally representative surveys provide valuable information about users’ willingness to accept and adopt AVs, individuals should be directly exposed to this emerging technology, to inform their perceptions of the capabilities and limitations of automation [5,6,13,22,23,24,25,26,27,28]. While we hope that automation solves our current transportation problems, we want users to have realistic expectations of automation and to calibrate their trust, perceptions, and hesitation with the actual capabilities of the systems. For this reason, we allowed riders to observe the safety operator during the shuttle ride and displayed the state of the automation (i.e., automation engaged or manual takeover by the safety operator) throughout the drive. It is conceivable that being able to observe the safety operator influenced our study results and can be explored in future study design. Most demonstration projects use low-speed shuttles that operate on small loops or a relatively short trip (≈2 miles), which also has its limitations of the potential usefulness of shuttles for older adults and those living in rural communities. Slow-speed shuttles were developed as a feeder or connecter of other modes of transit to compliment multimodal transportation (e.g., Mobility as a Service [MaaS]) and address the first-mile/last-mile problem (i.e., getting from your house to another mode of transit or from the train station to your final destination). Given the desire to improve accessibility and availability for those that are transportation disadvantaged or have mobility impairments, multimodal transportation may cause additional problems if adults are required to transfer to and from multiple vehicles on their trip. The strengths of the approach used in the current study include the use of an autonomous shuttle that is accessible (i.e., ADA compliant), operates on the highway and across various infrastructure and road types (e.g., gravel), interacts with mixed traffic and vulnerable road users, and would not require riders to use multiple modes of transportation to get to their final destination.

## Figures and Tables

**Figure 1 geriatrics-07-00140-f001:**
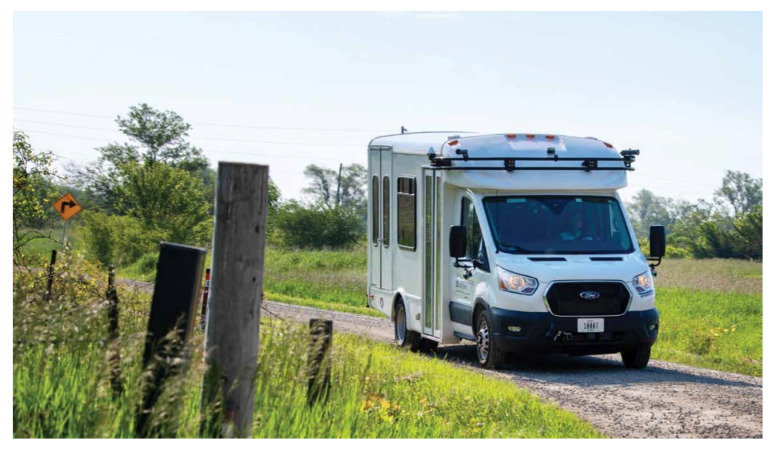
ADS Shuttle.

**Figure 2 geriatrics-07-00140-f002:**
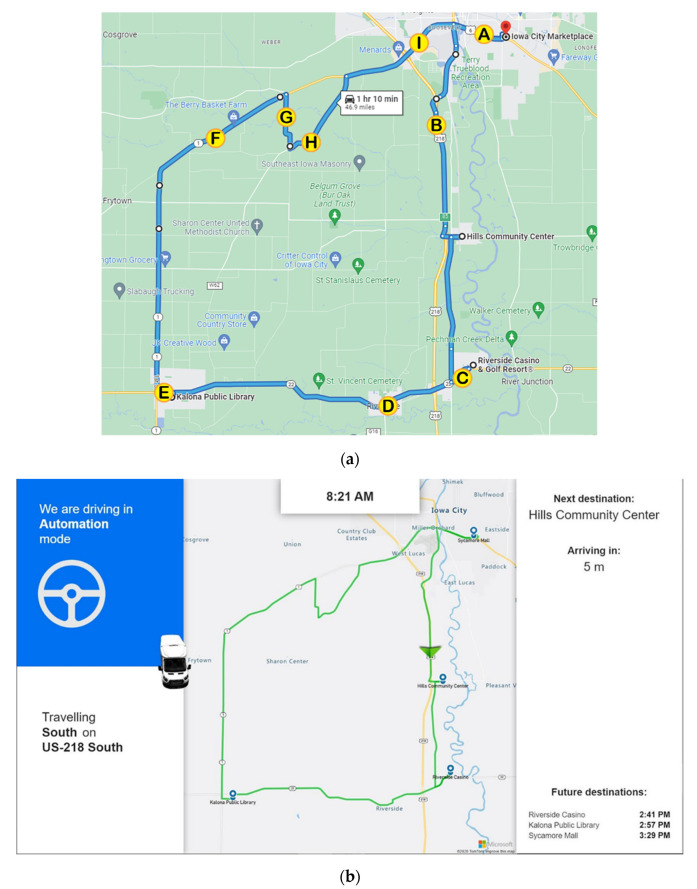
(**a**) Map indicating rating locations of anxiety; (**b**) Interactive map that was provided to the participant during their shuttle ride.

**Figure 3 geriatrics-07-00140-f003:**
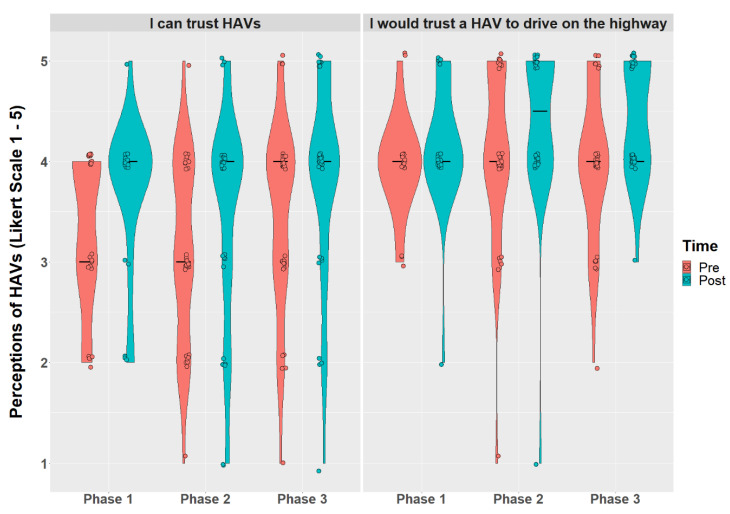
Violin plot with jitter and median line displaying trust and situational trust before and after riding in the shuttle between phases.

**Figure 4 geriatrics-07-00140-f004:**
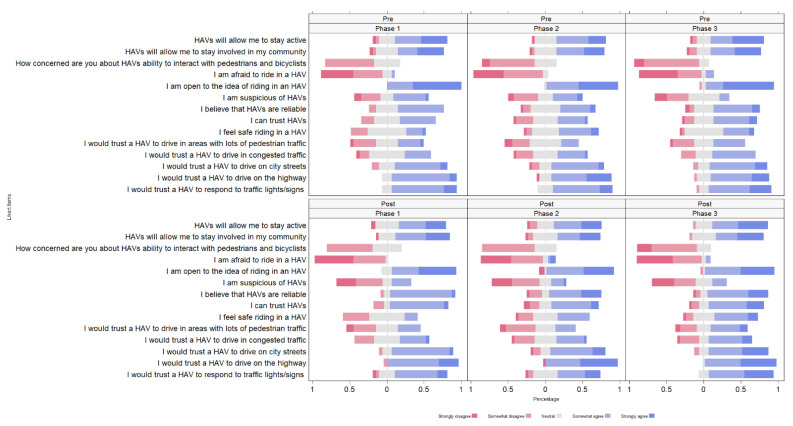
Diverging bar chart displaying the perceptions of HAVs before and after riding in the shuttle between phases.

**Figure 5 geriatrics-07-00140-f005:**
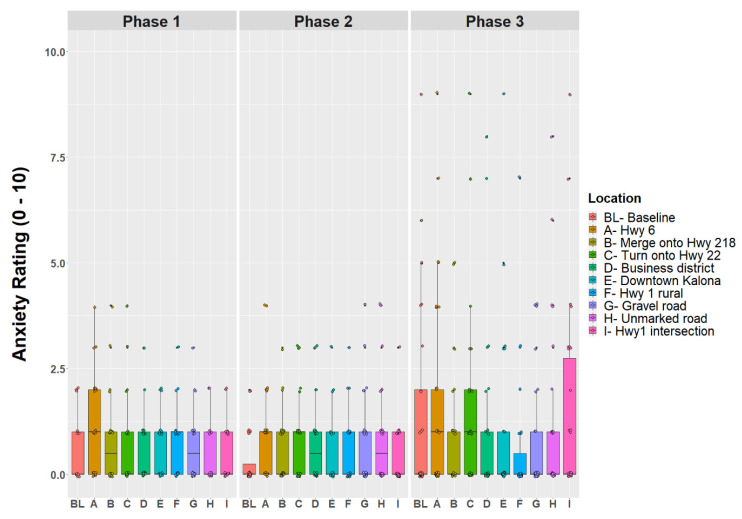
Boxplot with jitter displaying anxiety ratings by location and between the phases.

**Table 1 geriatrics-07-00140-t001:** Demographics and transportation preferences by phase.

Variable	Phase 1	Phase 2	Phase 3	Total Sample
	(n = 24)	(n = 30)	(n = 31)	(N = 85)
**% of drive in automation**				
Actual	58%	59%	93%	-
Perceived	44% ± 13%	48% ± 13%	76% ± 14%	57% ± 21%
**Age**				
65–74	16 (67%)	23 (77%)	16 (52%)	55 (65%)
75+	8 (33%)	7 (23%)	15 (48%)	30 (35%)
**Gender**				
Female	14 (58%)	14 (47%)	16 (52%)	43 (51%)
Male	10 (42%)	16 (53%)	15 (48%)	42 (49%)
**Impairment**				
Visual	2 (8%)	3 (10%)	2 (6%)	7 (8%)
Walking	3 (12%)	5 (17%)	5 (16%)	13 (15%)
Mobility device	1 (4%)	2 (7%)	3 (10%)	6 (7%)
None	19 (79%)	23 (77%)	25 (81%)	67 (79%)
**Education**				
Graduate degree	7 (29%)	11 (37%)	16 (52%)	34 (40%)
College degree	14 (58%)	12 (40%)	11 (35%)	37 (44%)
High school diploma	3 (13%)	7 (23%)	4 (13%)	14 (16%)
**Driving Restrictions**				
Yes (eyeglasses)	13 (54%)	21 (70%)	15 (48%)	49 (58%)
No	10 (42%)	9 (30%)	16 (52%)	35 (41%)
No license	1 (4%)	0 (0%)	0 (0%)	1 (1%)
**Annual Mileage**	1 (4%)			
<2000 mi	9 (38%)	2 (7%)	3 (10%)	6 (7%)
2000–6000 mi	6 (18%)	8 (27%)	8 (26%)	25 (29%)
6000–12,000 mi	8 (24%)	10 (33%)	14 (45%)	30 (35%)
>12,000 mi		10 (33%)	6 (19%)	24 (28%)
**Vehicle has ACC or LKA**				
No	15 (62%)	17 (57%)	19 (61%)	51 (60%)
Yes	9 (38%)	11 (37%)	12 (39%)	32 (38%)
Unsure	0 (0%)	2 (6%)	0 (0%)	2 (2%)

Note. Data are presented as count (n) and percentage (%).

**Table 2 geriatrics-07-00140-t002:** Perceptions of HAVs before and after riding in the AV by phase.

Item	Phase 1		Phase 2		Phase 3		Total Sample	
	(n = 24)		(n = 30)		(n = 31)		(N = 85)	
	Pre	Post	Pre	Post	Pre	Post	Pre	Post
How concerned are you about the HAVs ability to interact with pedestrians and bicyclists	2 (1)	2 (1)	2 (1)	2 (1)	2 (0)	2 (0)	2 (.25)	2 (1)
I would trust a HAV to drive on the highway	4 (0.25)	4 (0.25)	4 (1)	4.5 (1)	4 (0)	4 (1)	4 (0)	4 (1)
I am open to the idea of riding in an HAV	5 (1)	5 (1)	5 (1)	4 (1)	5 (1)	4 (1)	5 (1)	4 (1)
I would trust a HAV to respond to traffic lights/signs	4 (0)	4 (1)	4 (0)	4 (1)	4 (1)	4 (1)	4 (0)	4 (1)
I am afraid to ride in a HAV	2 (1)	1 (1)	2 (1)	2 (1)	1 (1)	2 (1)	2 (1)	2 (1)
I would trust a HAV to drive in areas with lots of pedestrian traffic	3 (2)	3 (2)	3 (1)	3 (1.75)	3 (2)	3 (2)	3 (2)	3 (2)
I would trust a HAV to drive on city streets	4 (0)	4 (0)	4 (1)	4 (.75)	4 (0)	4 (1)	4 (1)	4 (0)
I would trust a HAV to drive in congested traffic	3 (1.25)	3 (1.25)	3 (1.75)	3 (2)	4 (1)	4 (2)	3 (1)	3 (2)
I feel safe riding in a HAV	3 (.5)	3 (1)	4 (1)	3 (1)	3 (1)	4 (1)	3 (1)	3 (1)
I can trust HAVs	3 (1)	4 (0)	3 (1.75)	4 (1)	4 (1)	4 (.5)	3 (1)	4 (1)
I believe that HAVs are reliable	4 (1)	4 (0)	4 (1.75)	4 (1.75)	4 (1)	4 (.5)	4 (1)	4 (0)
I am suspicious of HAVs	3 (2)	2 (2.25)	3 (2)	2 (1.75)	3 (1)	2 (2)	3 (2)	2 (2)
HAVs will allow me to stay active	4 (2)	4 (2)	4 (1)	4 (1.75)	4 (2)	4 (1.5)	4 (2)	4 (2)
HAVs will allow me to stay involved in my community	4 (1.5)	4 (1.5)	4 (1.75)	4 (1.75)	4 (2)	4 (2)	4 (2)	4 (2)

Note. Data are presented as Median (Interquartile Range). Item responses were Likert-scaled ranging from 1 to 5 (strongly disagree to strongly agree).

## Data Availability

Data will be made publicly available for analysis and will be augmented for ease of use. It will be available for five years after completion of the study. Data can be accessed at https://data.adsforruralamerica.uiowa.edu/, accessed on 30 July 2022.

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
