# Peer review of "Autonomous Shuttle Operating on Highways and Gravel Roads in Rural America: A Demonstration Study"

_geriatrics, 2022, doi:10.3390/geriatrics7060140_

Round 1

Reviewer 1 Report

The authors assessed the views of older adults who took part in a demonstration project by riding on an autonomous shuttle in rural areas. The older adults provided ratings of their anxiety before and after the ride on various road types and vehicle speeds. The ratings showed that in general, older adults experienced increased trust and reduced suspicion after riding in the HAV.

This work adds to the sparse/limited literature regarding exposure to autonomous vehicles and how they are perceived, particularly by vulnerable populations, e.g., the elderly. The demonstration project is described well. 

Limitations were acknowledged. Figure 4 is difficult to read. I am not certain how this can be modified other than making it larger for the reader to see. 

The article is suitable for Geriatrics and expands upon the work done and published by Siegfried, A.L., et al (2021) in Geriatrics 6(2):47. 

Author Response

We appreciate your time, consideration, and expertise. We've responded below to your feedback and enclosed a document that contains all reviewer feedback and our responses to the reviewers.

Reviewer Comment: Limitations were acknowledged. Figure 4 is difficult to read. I am not certain how this can be modified other than making it larger for the reader to see. 

Response: We have enhanced all figures in this manuscript. Figures 3 and 4 required the most revisions but we now believe the figures are suitable for publication and will be clear to the reader.

Reviewer 2 Report

This paper has some interest to it, as there are three phases in a relatively stable context for the measures to take place.

However the paper needs quite a lot of improvement before it can be considered as publishable.  Some of the main issues are below.

Firstly there is little about how the subjects were recruited, including how many were circulated, how many replied, how many were deemed to be able to be subjects, and their interest which might [as the authors suggest] may already contain a positive bias. The paper is wek on the whole aspect of the subjects. More could be said about their recent driving history and its relation to automation [for example did their own vehicles have parking-assist technology or cameras, feedback systems, etc etc.].  

More needs to be said about the line of questions chosen and those rejected, or not considered, for the investigation. For example, how close they are to getting a new vehicle, whether the next one might have more HAV tec, etc- did they start with a battery of qns then reduce them, or what? How were the questions constructed and chosen. 

They need to explain more about whether there are any steps in the vehicle, where shopping or luggage might go, how fast or slow for access and egress, etc. etc

The use of 5-point Likert scales means the findings are weaker than they would have been with 6 or 7 point scales. Some people are notorious for omitting extremes which renders a 5-point scale into effectively a 3-point scale, which makes the ANOVAs less valid as they rely on interval measurement [the Likert scale is ordinal], normal distributions etc. Indeed the data are unsurprisingly not normally distributed.   

The presentation of the findings is a big problem. Table 1 for example, shows 85% as aged 65 or over- presumably some would not give their ages or gender. Given that the 85 are distributed over three phases, so there are very few in each phase. Added to the limits of 5-point scales, many analyses are based on very few subjects.  If they are all bundled together in the analyses, this needs to be stated, by providing the Ns with the reported statistics. 

Also on presentation, Figure 3 is difficult to follow and too small and the colours are difficult to see and what the figure is meant to be showing is not clear from simply looking at it. Figure 4 I am afraid is really awful to look at and interpret and some alternative presentation is needed for both these figures. They really look terrible. Table 2- what on the first line do the figures 2(1), 2(1), etc mean?  Same for all the lines, the table needs to tell us exactly what is being given by way of the findings, and I am not sure why they chose medians instead of means.  Really not helpful. Figure 5 is poorly presented and by that time I gave up on Figure 6.  So a major rethink of presentation across the board. 

The discussion is OK as far as it goes, but is weak in many ways.  So some improvement in the sense of less suspiciousness. Then limitations of the sample, and I do hope they have a better chosen and much larger sample for the next three phases, and preferably longer scaling as well. All the possibilities here for future adoption of AVs, either owned or on a circuit like this, not to mention all the legal issues about what happens if an AV runs into a shopping trolley or a buggy, etc.  More importantly Issues such as on-demand [like MaaS for example] AVs, help with lifting and carrying, how the timetabling would work out if several using wheelchairs, etc.  And what about small towns and less rural implications? Loads of logistics need to be considered, and at least those could have gone into a conclusion/ implications/recommendations section at the end. 

I could say more, and am happy to extend my recommendations if that would help.  The paper is not publishable as it stands.

Author Response

We appreciate your time, consideration, and expertise. We've responded below to your feedback and enclosed a document that contains all reviewer feedback and our responses to the reviewers.

Reviewer Comment: Firstly there is little about how the subjects were recruited, including how many were circulated, how many replied, how many were deemed to be able to be subjects, and their interest which might [as the authors suggest] may already contain a positive bias. The paper is wek on the whole aspect of the subjects. More could be said about their recent driving history and its relation to automation [for example did their own vehicles have parking-assist technology or cameras, feedback systems, etc etc.].  

Response: More information has been provided regarding the study subjects and our inclusion and exclusion criteria:

“Adults over the age of 65 were recruited to ride inside the AV. Eligible participants were: a) able to sit for up to 3 hours at a time; b) felt comfortable riding in the shuttle with strangers; and c) able to read English. Participants were not eligible to participate in the study if they were: a) having difficulties with memory or confusion; b) diagnosed with dementia, delirium, mild cognitive impairment, or other severe neurological impairments (i.e., Parkinson’s, Huntington’s); c) diagnosed with an anxi-ety disorder accompanied with an acute anxiety or a panic attack in the last 6 months; d) diagnosed with seizures, narcolepsy, or epilepsy with an episode in the past 12 months; e) deaf in both ears; and f) prone to motion sickness. They were recruited from the communities along the route using flyers and word of mouth as well as through emails sent through the National Advanced Driving Simulator registry. Recruitment emails were sent to 701 older adults in the registry. 235 adults completed an online survey from the registry, flyers, or word of mouth. Of the 235 adults, 189 individuals met the inclusion and exclusion criteria. 105 participants were called sequentially and 85 were scheduled to participate in the study. 20 of the 105 adults had scheduling conflicts and expressed interest in participating in a future phase of the project.”

Reviewer Comment: More needs to be said about the line of questions chosen and those rejected, or not considered, for the investigation. For example, how close they are to getting a new vehicle, whether the next one might have more HAV tec, etc- did they start with a battery of qns then reduce them, or what? How were the questions constructed and chosen. 

Response: Although it is interesting to explore the relationship between adults’ owning or being willing to purchase a vehicle with lower levels of automation and their perceptions of highly autonomous vehicles, it is beyond the scope of this project. This would lead to a different manuscript, with different RQs, and different analyses. We can certainly explore this RQ once we have a larger dataset and have completed all phases of the demonstration project.

This recent manuscript (10.3389/fpsyg.2021.682973) is the closest fit I could find to exploring older adults’ preferences of different levels of automation. The largest weakness in these survey studies, is they do not expose the individuals to the technology which limits participants’ understanding of the systems and the current capabilities/limitations of automation. We sought to expose older adults directly to the autonomous vehicle so that they could better understand the current state of the technology. We will solicit older adults’ perceptions of their current Level 1 or Level 2 automation system in the next few phases of our project. We appreciate your suggestion as this is certainly of interest.

In regards to the items we analyzed, additional rationale has been provided as to how the items were selected: “The survey contained 34 items but only 14 items were analyzed in this study as they were particularly relevant to the automation capabilities that changed throughout these specific phases (i.e., Phases 1 through 3). Furthermore, only a portion of the items were analyzed to limit the potential of type I error.” Additionally, we detailed our approach to compare for multiple comparisons at the end of the data analysis paragraph on Page 5.

Reviewer Comment: They need to explain more about whether there are any steps in the vehicle, where shopping or luggage might go, how fast or slow for access and egress, etc. etc

Response: More information has been provided regarding ingress and egress: “To support accessibility needs of older adults, the vehicle has storage racks for luggage or groceries and a low first entry step, deep step wells, and a ramp that is compliant with the Americans with Disabilities Act (ADA) and Rehabilitation Act.”

Reviewer Comment: The use of 5-point Likert scales means the findings are weaker than they would have been with 6 or 7 point scales. Some people are notorious for omitting extremes which renders a 5-point scale into effectively a 3-point scale, which makes the ANOVAs less valid as they rely on interval measurement [the Likert scale is ordinal], normal distributions etc. Indeed the data are unsurprisingly not normally distributed.   

Response: The measurement literature pertaining to 5- vs 7-point Likert scales is equivocal. 5-point Likert scales often display higher reliability but this is also dependent on the amount of survey items, participant demographics, mode of data collection, etc.. We chose to use a 5-point Likert scale as it has been shown to increase response rate and quality while also reducing participants’ frustration. (PMID: 1737708).

Regarding the analysis approach, ANOVAs and most other parametric statistics (i.e., based on central tendencies) are robust and can handle small sample sizes, unequal variances, and non-normal distributions. (https://doi.org/10.1111/j.1365-2923.2008.03172.x). However, we tried to balance our approach by using a nonparametric post hoc analyses.  

Reviewer Comment: The presentation of the findings is a big problem. Table 1 for example, shows 85% as aged 65 or over- presumably some would not give their ages or gender. Given that the 85 are distributed over three phases, so there are very few in each phase. Added to the limits of 5-point scales, many analyses are based on very few subjects.  If they are all bundled together in the analyses, this needs to be stated, by providing the Ns with the reported statistics. 

Response: We are grateful that you caught this error. We intended to display the results in table 1 as count (n) and frequency (%). We provided the count but left out the frequency which is why “(%)” was displayed after each count. The values did not change but the frequencies are now provided in the table. We’ve made these changes as track changes within the table.

Reviewer Comment: Also on presentation, Figure 3 is difficult to follow and too small and the colours are difficult to see and what the figure is meant to be showing is not clear from simply looking at it. Figure 4 I am afraid is really awful to look at and interpret and some alternative presentation is needed for both these figures. They really look terrible. Table 2- what on the first line do the figures 2(1), 2(1), etc mean?  Same for all the lines, the table needs to tell us exactly what is being given by way of the findings, and I am not sure why they chose medians instead of means.  Really not helpful. Figure 5 is poorly presented and by that time I gave up on Figure 6.  So a major rethink of presentation across the board. 

Response: Thank you for your constructive feedback. Figures 3 and 4 have been modified to improve the clarity and resolution of the figures.

Table 2 displays the medians and interquartile range (IQR = Quartile 3 – Quartile 1; 75-25 = 50% of the data) which is the most suitable approach to display data that are “unsurprisingly not normally distributed”. How data were displayed in Table 2 is noted at the bottom of Table 2. The display of means for central tendency is often used incorrectly, especially when the data is skewed. Therefore, we displayed the data as median and IQR.

Figures 5 and 6 were also modified to enhance clarity (Font was enlarged and bolded). Figure 5 displays anxiety ratings across time (i.e., from baseline to throughout the shuttle ride) between the phases. Figure 6 displays anxiety ratings at different locations of the drive and displays each individual's rating. Ratings are displayed as jitters to show that central tendencies (i.e., median or mode) may hide individuals that report high anxiety levels at various locations of the drive. To aid the reader, we’ve added this description of Figure 6: “Individual ratings of anxiety are also displayed within Figure 6. Descriptively, older adults in Phase 3 reported higher ratings of anxiety. The violin plots in Figure 6 display the density of older adults’ anxiety ratings as well as the distribution of individuals’ anxiety ratings.” Lastly, we revised the titles of the figures.

Reviewer Comment: The discussion is OK as far as it goes, but is weak in many ways.  So some improvement in the sense of less suspiciousness. Then limitations of the sample, and I do hope they have a better chosen and much larger sample for the next three phases, and preferably longer scaling as well. All the possibilities here for future adoption of AVs, either owned or on a circuit like this, not to mention all the legal issues about what happens if an AV runs into a shopping trolley or a buggy, etc.  More importantly Issues such as on-demand [like MaaS for example] AVs, help with lifting and carrying, how the timetabling would work out if several using wheelchairs, etc.  And what about small towns and less rural implications? Loads of logistics need to be considered, and at least those could have gone into a conclusion/ implications/recommendations section at the end. 

Response: We’ve added more information regarding our study design: “For this reason, we allowed riders to observe the safety operator during the shuttle ride and displayed the state of the automation (i.e., automation engaged or manual takeover by the safety operator) throughout the drive. It is conceivable that being able to observe the safety operator influenced our study results and can be explored in future study design.”

MaaS is now mentioned in the last paragraph of the discussion section: “Slow-speed shuttles were developed as a feeder or connecter of other modes of transit to compliment multimodal transportation (e.g., Mobility as a Service [MaaS]) and address the first-mile/last-mile problem (i.e., getting from your house to another mode of transit or from the train station to your final destination).”

Thank you for your suggestions. We’ve added additional next steps in the discussion section: “To overcome sampling bias, future phases of this project may target older adults that are reluctant to ride in an autonomous shuttle, persons with disabilities that require assistance with ingress or egress, or those with weak mental models (i.e., understanding) of automation. Results from future demonstration projects may elucidate the dynamic effects of the regulation, policy, and availability of AVs on older adults’ perceptions of HAVs.”

Reviewer 3 Report

The authors analyse trust and anxiety levels of older adults using AV's in different road settings, with different degrees of automation. 
Unfortunatley the drivers do not take into account the fact that there was a safety-driver behind the wheel at all times. Thus results might not be as reliable as suggested. 

It is not clear if these vehicles are intended to be used with safety driver (but then the question about the benefits of the AV arises) or if in future these vehicles should run as autonomous shuttles (maybe just with attendant staff). In the second case, I would like to invite the authors to add a survey item asking about the hypothetical scenario: no safety driver behind the wheel. -> how would trust and anxiety levels change? 

Figures are difficult to read due to small font, poor resolution etc.

Author Response

We appreciate your time, consideration, and expertise. We've responded below to your feedback and enclosed a document that contains all reviewer feedback and our responses to the reviewers.

Reviewer Comment: The authors analyse trust and anxiety levels of older adults using AV's in different road settings, with different degrees of automation. Unfortunately, the drivers do not take into account the fact that there was a safety-driver behind the wheel at all times. Thus, results might not be as reliable as suggested. 

Response: During study design, we had to decide between being transparent or deceptive without research participants (i.e., riders). Given that emerging technologies are constantly improving, it is important to show the current state of the technology (i.e., automation). The participants were able to actively observe the driver and we displayed the vehicle automation state (i.e., driving manually or in automation) on the heads-up display (i.e., monitor at the front of the shuttle as depicted in Figure 2b). We understand that this is a limitation and have further detailed this in our discussion section. Speculation may suggest that we could have had different results based on our design (i.e., using a curtain to prevent the riders from observing the driver). In the US, the National Highway Traffic Safety Administration approves the shuttle route and requires a safety operator to have hands on the vehicle controls at all times. Sometimes this looks a bit different in the slow-speed shuttles which use a controller that resembles an Xbox controller.

We’ve added more information regarding our study design in the discussion section: “For this reason, we allowed riders to observe the safety operator during the shuttle ride and displayed the state of the automation (i.e., automation engaged or manual takeover by the safety operator) throughout the drive. It is conceivable that being able to observe the safety operator influenced our study results and can be explored in future study design.”

Reviewer Comment: It is not clear if these vehicles are intended to be used with safety driver (but then the question about the benefits of the AV arises) or if in future these vehicles should run as autonomous shuttles (maybe just with attendant staff). In the second case, I would like to invite the authors to add a survey item asking about the hypothetical scenario: no safety driver behind the wheel. -> how would trust and anxiety levels change? 

Response: As mentioned above regarding NHTSA, policy will dictate how this technology is utilized in public or private transit. As of now, a driver must be behind the ‘wheel’ in the US. Discussions in the AV field suggest that the driver may be reutilized and trained to assist passengers with transfer, ingress, egress, and route planning. However, this is yet to come to fruition. You raise a great point and offer a great survey item that we will add to future phases of this demonstration project.

Reviewer Comment: Figures are difficult to read due to small font, poor resolution etc.

Response: We completely agree and have revamped all figures. Specifically, all figures now meet resolution requirements and have enhanced clarity. Figures 3 and 4 required the most revisions and have also been thoroughly modified and improved.

Round 2

Reviewer 2 Report

Most of the changes I recommended have not been implemented, especiaally those relating to the presentation and discussion.   Therefore I cannot recommend acceptance of the paper. 

Author Response

Thank you for your time and willingness to participate in the second round of revisions. We believe we have addressed your first round of revisions as detailed by our itemized responses. The revisions were reflected in our revised manuscript with all edits occurring in track changes per author guidelines. Specifically, we have modified all of our figures and added substance to our discussion section to align with your feedback. Is it possible that your MS Office options did not properly display our track changes (or the system provided you with our original submission rather than the revised manuscript)?

Reviewer 3 Report

Thanks for taking your time to revise the manuscript. I think it would be good to add some of your explanations that you provided in the response letter directly in the manuscript. E.g. 

In the US, the National Highway Traffic Safety Administration approves the shuttle route and requires a safety operator to have hands on the vehicle controls at all times. Sometimes this looks a bit different in the slow-speed shuttles which use a controller that resembles an Xbox controller.

Graphics are still difficult to read, especially the two violin plots. Is there a better way/ graph type to plot Figure 6 (or, make the figure wider and place the legend inside the plot, there is enough space for it.? 
Also figure 5 seems unnecessary. 

Some typos in Table 1 (percentages)

Author Response

Reviewer: Thanks for taking your time to revise the manuscript. I think it would be good to add some of your explanations that you provided in the response letter directly in the manuscript. E.g. 

  • In the US, the National Highway Traffic Safety Administration approves the shuttle route and requires a safety operator to have hands on the vehicle controls at all times. Sometimes this looks a bit different in the slow-speed shuttles which use a controller that resembles an Xbox controller.

Response to Reviewer: Thank you for your time, expertise, and patience! We truly appreciate your willingness to provide a second round of feedback. We've incorporated your suggestion and added this information in the discussion section as it nicely flows in our limitations and future work section:

"The presence of a safety operator in the driving seat likely influenced older adults’ perceptions while riding in the shuttle. In the US, the National Highway Traffic Safety Administration approves the proposed shuttle route and requires safety operators to always have their hands on the vehicle controls. This often looks different in the slow-speed shuttles (e.g., Navya Autonom or EasyMile EZ10), which use a remote controller resembling an Xbox controller or drone remote controller. Results from future demonstration projects may elucidate the effects of changes to regulation, policy, and availability of AVs on older adults’ perceptions of HAVs."

Reviewer: Graphics are still difficult to read, especially the two violin plots. Is there a better way/ graph type to plot Figure 6 (or, make the figure wider and place the legend inside the plot, there is enough space for it.? 
Also figure 5 seems unnecessary. 

Response to Reviewer: Per your suggestion, Figure 5 has been removed. Figure 6 was revised (which is now Figure 5) by adding grid lines, switching from violin plot to boxplot, using color fill rather than outlines, and bolded and increased the font size for the titles of our y-axis, x-axis, and facets. 

Reviewer: Some typos in Table 1 (percentages)

Response to Reviewer: Thank you for catching this error within our track changes. We've removed the typos from our table (1 value had a missing percentage sign and 2 values had repeating digits).